# Risk Factors of In-Hospital Mortality in Non-Specialized Tertiary Center Repurposed for Medical Care to COVID-19 Patients in Russia

**DOI:** 10.3390/diagnostics11091687

**Published:** 2021-09-15

**Authors:** Anton Kondakov, Alexander Berdalin, Vladimir Lelyuk, Ilya Gubskiy, Denis Golovin

**Affiliations:** Radiology and Clinical Physiology Research Center, Federal Center of Brain Research and Neurotechnologies of the Federal Medical Biological Agency, 117997 Moscow, Russia; berdalin@fccps.ru (A.B.); vglelyuk@fccps.ru (V.L.); gubskiy.i@fccps.ru (I.G.); golovin@fccps.ru (D.G.)

**Keywords:** COVID-19, risk factors, arterial hypertension, computed tomography, body mass index

## Abstract

The purpose of our study is to investigate the risk factors of in-hospital mortality among patients who were admitted in an emergency setting to a non-specialized tertiary center during the first peak of coronavirus disease in Moscow in 2020. The Federal Center of Brain and Neurotechnologies of the Federal Medical and Biological Agency of Russia was repurposed for medical care for COVID-19 patients from 6th of April to 16th of June 2020 and admitted the patients who were transported by an ambulance with severe disease. In our study, we analyzed the data of 635 hospitalized patients aged 59.1 ± 15.1 years. The data included epidemiologic and demographic characteristics, laboratory, echocardiographic and radiographic findings, comorbidities, and complications of the COVID-19, developed during the hospital stay. Results of our study support previous reports that risk factors of mortality among hospitalized patients are older age, male gender (OR 1.91, 95% CI 1.03–3.52), previous myocardial infarction (OR 3.15, 95% CI 1.47–6.73), previous acute cerebrovascular event (stroke, OR = 3.78, 95% CI 1.44–9.92), known oncological disease (OR = 3.39, 95% CI 1.39–8.26), and alcohol abuse (OR 6.98, 95% CI 1.62–30.13). According to the data collected, high body mass index and smoking did not influence the clinical outcome. Arterial hypertension was found to be protective against in-hospital mortality in patients with coronavirus pneumonia in the older age group. The neutrophil-to-lymphocyte ratio showed a significant increase in those patients who died during the hospitalization, and the borderline was found to be 2.5. CT pattern of “crazy paving” was more prevalent in those patients who died since their first CT scan, and it was a 4-fold increase in the risk of death in case of aortic and coronal calcinosis (4.22, 95% CI 2.13–8.40). Results largely support data from other studies and emphasize that some factors play a major role in patients’ stratification and medical care provided to them.

## 1. Introduction

### 1.1. Background

COVID-19 became the first pandemic infection in the 21st century. As of 31st of August 2021, 215,714,824 people were infected with SARS-CoV-2 virus and 4,490,753 have died. Despite the high priority of response measures, established by most of the countries, SARS-CoV-2 still is spreading in different regions, with a tendency to increase the number of infected [1].

Coronaviruses are enveloped, single-stranded RNA viruses, that are commonly found in people and other mammals and cause respiratory, gastrointestinal, and neurologic diseases. The most common coronaviruses cause symptoms of the common cold in the immunocompetent population. SARS-CoV-2 is the third coronavirus to cause severe disease and spread all over the world in the last two decades. The first one was the virus of severe acute respiratory syndrome (SARS) that appeared in Foshan in China and caused the SARS-CoV pandemic 2002–2003. The next one was called Middle Eastern respiratory syndrome (MERS), which appeared on the Arabian Peninsula in 2012. Current SARS-CoV-2 virions have a diameter from 60 to 140 nm and characteristic spikes with a length of 9–12 nm, making them look like a stellar corona [2].

SARS-CoV-2 has been found to have a higher transmission rate, compared to SARS and MERS. Its basic reproduction number (R0) was previously reported to be 3.32 (95% CI 3.24–3.39), and the incubation period is about 5.24 days (95% CI: 3.97–6.50), with variation in time of symptom appearance across countries [3]. In other studies, basic reproduction number has been found to be 2.21 (95% CI 1.86–2.63) in Western Europe [4], 5.9 (95% CI 4.7–7.5) in the USA [5], and 4.08 (95% CI 3.09–5.39) at the global level [6].

In the earliest stage, SARS-CoV-2 is targeted to nasal cavities’ epithelium, bronchial epithelium, and alveolocytes, which is realized by the binding of a virus’ S-protein to a receptor of angiotensin-converting enzyme type 2 (ACE2) on the cellular surface. Although it was thought that an increase in the number of ACE2, caused by ACE inhibitors and angiotensin receptor blockers (ARB), increases susceptibility to the infection, large groups of investigators found no link between these medications and risks of infection or in-hospital mortality. For example, in a Dutch research on 4480 patients with COVID-19, previous treatment with inhibitors of ACE and ARB has not influenced the mortality rate [7]. Research on comparison of ACE inhibitors and ARB with a frequency of positive SARS-CoV-2 polymerase chain reaction tests among 18,472 patients resulted in no significant association between them and supported the practice of treatment with these types of medications during the COVID pandemic [8].

In the later stages of the infection process, when virus replication speeds up, the integrity of the epithelial-endothelial barrier becomes weaker. Besides epithelial cells, SARS-CoV-2 infects endothelial cells of lung capillaries, which leads to an increase in inflammatory reaction and influx of monocytes and neutrophils. The autopsy on these stages of the disease shows diffuse thickening of an alveolar wall with mononuclear and macrophagic infiltration of airspaces, with endothelialitis. During the infectious process, the interstitial inflammatory infiltrates and concurrent edema are visualized by CT as ground-glass opacities (GGO). The evolution of the infiltration process leads to lung edema, which fills up alveolar airspace and forming of hyaline membranes, which corresponds to the early stage of acute respiratory distress syndrome [2,3].

In summary, endothelial barrier impairment, alveolar-to-capillary oxygen transmission, and oxygen diffusion are characteristic parts of COVID pneumonia. Along with that, COVID-19 may cause lymphopenia by infecting and causing necrosis of T-lymphocytes.

Classical clinical symptoms of COVID-19 are fever (38.1–39.0 °C), cough, and fatigue, and most often complication is acute respiratory distress syndrome (ARDS). It had been found that fever over 39.0 °C, shortness of breath, and anorexia are more often in critically ill patients [3].

In severe cases of COVID-19 rapid activation of coagulation and consumption of blood coagulation factors occurs [9]. It is recognized that the pathophysiology of coagulation disturbances in COVID-19 mostly relies on some complicated interactions between proinflammatory cytokines, platelet activation, and downregulation of natural anti-coagulation system on the one side and endothelial dysfunction, von-Willebrand factor secretion, and complement activation on the other [10]. Inflamed lung parenchyma and damaged lung endothelium may cause the microthrombi formation and lead to a higher frequency of thrombotic complications, such as deep vein thrombosis, pulmonary embolism, and arterial thrombosis (e.g., in case of ischemic infarction and limb ischemia) in severely ill patients. To date, it had been found that coagulopathy in the case of coronavirus infection has characteristics of both disseminated intravascular coagulation (DIC) syndrome and local thrombotic events [11,12]. It is also reported that in some cases, due to imbalance in platelet production and consumption, some patients have an increased risk of bleeding complications but this condition is thought to be rare [10].

### 1.2. Risk Factors of Lethality and Severe Course of COVID-19

The combination of different COVID-19 patient characteristics leads to a different course of the disease that varies from asymptomatic to severe and may cause need in respiratory assistance and even death [13,14]. Thus, identification of risk factors of severe course of the disease is very valuable for timely hospital admission and precise patient status monitoring.

For the identification of these risk factors many studies were performed, a short review of their main results is presented below. Lack of standard study protocol, different inclusion criteria, and various treatment strategies result in different risk factors. This can be overcome by a unified protocol of data collection and analysis which has yet to be developed.

In a systematic review, Yewei Xie et al. showed that one of the risk factors of a COVID-19 patient’s death is the patient’s gender. In the overall analysis, males have a 35% higher chance of death compared to women (fatality rate ratio was found to be 1.35, with a 95% CI 1.35–1.35), and this tendency was shown in all regions and age groups. Moreover, the fatality rate ratio for males to females is higher in the younger age group (2.74, 95% CI 2.66–2.82, in the age group younger than 60 years) compared to older patients (1.83, 95% CI 1.82–1.83, in the age group 60 years and older) [3]. Similar results were found in the meta-analysis by Vignesh Chidambaram et al. [15], with a fatality rate ratio of 1.45 (95% CI 1.23–1.71). Despite few conflicting results [16], the male gender is recognized as a major risk factor of death [17] and risk factor of ICU admission [18] in other studies.

Age is an undeniable risk factor of severe course and death in coronavirus infection, with an increase in the probability of death in the older population [3,15,17,19,20]. Older age is an independent risk factor of severe course and mortality in coronavirus pneumonia (in general, patients older than 75 years had a 13-fold increase of risk of death (95% CI 9.13–17.85) compared to patients younger than 65 years). This effect can be partially explained by interfering comorbidities, which are more common in the older population [21].

There are conflicting results about smoking status and the risk of mortality in COVID. In the early report of Constantine Vardavas and Katerina Nikitara, there was a negative influence of smoking status shown on the outcome of coronavirus pneumonia, based on limited data [22]. However, in a systematic review, Giuseppe Lippi and Brandon Michael Henry found no statistically significant difference among the risk of mortality for smokers and non-smokers (OR 1.69; 95% CI 0.41–6.92) [23]. In a recent review on a link between smoking status and risk of severe course of COVID-19 it was shown that smokers have a higher chance of critical illness (OR 1.58; 95% CI 1.16–2.15) and death (OR 1.35; 95% CI 1.12–1.62). Similarly, it was reported that former smokers have a higher risk of critical illness and death (OR 2.48; 95% CI 1.64–3.77 and 2.58; 95% CI 2.15–3.09, respectively) [24].

The influence of high body mass index (BMI) on severity and mortality is often mentioned in different articles. In a meta-analysis of 22 studies, it was found that BMI was higher in severe and critically ill patients compared to mild disease (average difference was 2.48 kg/m^2^, 95% CI 2.00–2.96 kg/m^2^). In addition, obesity in patients with COVID-19 was associated with negative outcomes (OR 1.683, 95% CI 1.408–2.011), which were defined as severe disease, the need for intensive care, the use of invasive mechanical ventilation, or disease progression. Obesity as a risk factor was more pronounced in younger patients (odds ratio 3.30 vs. 1.72). In this analysis, obesity did not increase the risk of hospital mortality (odds ratio 0.89, 95% CI 0.32–2.51) [25]. On the other hand, a review based on data from the AHA register of cardiovascular diseases showed that a high body mass index is a risk factor for patient death in a medical institution: the risk ratio for III obesity class was 1.26 (95% CI 1.00–1.58). The need for mechanical ventilation was also higher in overweight and obese patients I–III classes (odds ratio 1.28 [95%CI 1.09–1.51], 1.54 [1.29–1.84], 1.88 [1.52–2.32], and 2.08 [1.68–2.58], respectively) [26].

Comorbidities significantly worsen the patient’s prognosis. Already in the early stages of the pandemic, links between hypertension, cardiovascular disease, chronic kidney disease, and diabetes have been demonstrated [27]. These conclusions were confirmed by subsequent studies. In a recent systematic review and meta-analysis by Maryam Honardoost et al., it was shown that disease severity was significantly higher among patients with a history of cerebrovascular diseases (odds ratio 4.85, 95% CI: 3.11–7.57), cardiovascular diseases (OR 4.81 95% CI: 3.43–6.74), chronic lung diseases (4.19, 95% CI: 2.84–6.19), malignancies (OR 3.18, 95% CI: 2.09–4.82), diabetes (OR 2.61, 95% CI: 2.02–3.3), and arterial hypertension (OR 2.37, 95% CI: 1.80–3.13) [28].

Biochemical blood test abnormalities in patients with severe infection are noteworthy. Older adults were reported to have higher levels of low-density lipoproteins, a larger width of red blood cell distribution, and higher levels of the D-dimer than other age groups, suggesting that these patients were more likely to have impaired blood clotting function [29].

Another coagulation system status marker, platelet count, was also considered as an independent risk factor. It should be noted that according to the meta-analysis by G. Lippi et al., thrombocytopenia was a predictor of the development of severe disease, while the weighted average difference in the number of platelets between patients with mild and severe coronavirus infection was −31 × 10^9^/L (95% CI ranges from−35 to −29 × 10^9^/L), and when comparing patients with a mild course of the disease and deceased patients, the weighted average difference increases to −48 × 10^9^/L (95% CI ranges from −57 to −39 × 10^9^/L), and the risk ratio of severe disease in patients with thrombocytopenia was found to be 5.1 (95% CI 1.8–14.6) [30].

In addition to these parameters, it is necessary to include a widespread and easily accessible marker of the severity of the process—lymphopenia, which is accompanied by leukocytosis and neutrophilia [31]. Based on numerous data A.-P. Yang et al. suggested using the neutrophil-to-lymphocyte ratio (NLR) as a predictive marker of the severe course of the disease [32]. It was shown that high values of this marker are independently associated with a severe course of the disease (risk 2.46, 95% CI 1.98–4.57), and the threshold value is 3.3 [32]. The data were also confirmed in the paper by J. Liu, but the threshold value of this parameter for discriminating patients into high- and low-risk groups was reduced to 3.13, with 50% of patients in the higher NLR group suffering from critically severe disease [33].

In biochemical blood tests, the most significant predictive parameters of severe disease were elevated levels of C-reactive protein (weighted mean difference 42.7 mg/L, 95% CI 31.12–54.28), lactate dehydrogenase (weighted mean difference 137.4 U/L, 95% CI 105.5–169.3), procalcitonin (weighted mean difference 0.07 ng/mL, 95% CI 0.05–0.10), ALT (weighted mean difference 5.12 U/L, 95% CI 0.82–9.42), AST (weighted mean difference 8.51 U/L, 95% CI 5.01–12.01), and creatinine (weighted mean difference 4.57 µmol/L, 95% CI 0.64–8.50) [34].

CT analysis uses a semi-quantitative scale for assessing lung tissue involvement, according to which the degree of damage to each lobe of the lung is determined on a 5-point scale, then all points are summed up to get a total damage score, giving a total score ranging from 0 to 25 points [35].

According to the results of published studies, this summary assessment is the independent risk factor for a severe course and unfavorable outcome of the disease. At the same time, with a total score of more than 18, the risks of death were higher both in the univariate analysis (HR, 8.33; 95% CI, 3.19–21.73) and in the multivariate analysis (HR, 3.74; 95% CI, 1.10–12.77), that was done with 130 patients [36]. Similar data are presented in other studies (in patient cohorts ranging from 128 to 572 participants), both in semi-quantitative and quantitative assessment of lung tissue involvement [37].

When studying specific CT patterns of the disease, it was shown that the presence of mosaic perfusion is predictive for the development of a severe course of the disease, while the presence of an air bronchogram, the location and structure of areas of reduced pneumatization of lung tissue (ground-glass opacities—GGO), and the presence of thickened vascular bundles did not show such relationships [38]. According to N. Lassau et al., the peripheral location of sites of reduced pneumatization of the pulmonary parenchyma is a predictor of a milder course of infection, while the symptom of “crazy paving” (GGO and thickened interlobular septa) and the prevalence of the disease in the lungs are predisposing factors for the severe course of the disease [39].

The main findings of some of the aforementioned studies are summarized in Table 1.

In the Russian population, Youri Kirillov et al. previously reported their data based on retrospective cohort analysis of electronic medical records of 1487 deceased patients with RT-PCR confirmed COVID-19. According to their report, the median age for lethal outcome was 71 years of age among males and 78 for females. Arterial hypertension was reported in 81% of their cases, diabetes mellitus was found to be a comorbidity in 37% of cases. About 12% of deceased patients had cancer [40].

Thus, there are data on various risk factors for severe coronavirus infection. In this article, we present the results of our analysis of the intrahospital COVID-19 course collected during the work in March–July 2020. The presented data on the patient population will complement the knowledge about the disease and allow us to develop tactics for the distribution of patients, stratification of their risk level, and algorithms for maintenance therapy.

## 2. Materials and Methods

### 2.1. Patients

We analyzed the database containing a total of 635 patients who were treated at the Federal Center of Brain research and Neurotechnologies, repurposed for Health Care for COVID-19 in the spring of 2020. In total, more than 700 patients passed through our center, but some of them refused to be hospitalized, were transferred to other medical organizations, and some were not confirmed to have coronavirus pneumonia. Of those included in the analysis, a total of 53 patients (8.3%) died.

Among the patients included in the study, 351 (55.3%) men and 284 (44.7%) women were included. The mean age was 59.1 ± 15.1 years, including 57.0 ± 15.1 years for men and 61.8 ± 14.6 years for women.

Total of 56 patients reported smoking (8.8%—lower than recently published data on prevalence of current smoking in Russia [41]), while 10 patients (1.6%) did not complete this form in the questionnaire and were excluded from the analysis of smoking risk assessment. Arterial hypertension was detected in 318 patients (50.1%—corresponds to previously reported data about the general Russian population [42]), myocardial infarction—in 50 patients (7.9%), 25 patients had a stroke before (3.9%), COPD was found in 41 patients (6.5%), 18 patients suffered bronchial asthma (2.8%), 32 patients were admitted with known cancer after treatment (5%), 97 patients (15.3%) suffered from well-controlled diabetes mellitus (this is higher than in corresponding age group of the general Russian population that was reported before [43]). In a hospital setting, if necessary, correction of antihypertensive, hypoglycemic, protective, and other permanent therapy was performed.

Patients were transferred to our Center by ambulance on the basis of proximity of their location to the Center, and clinical condition (decreased blood oxygen saturation, severe symptoms of pneumonia) under supervision of Moscow Ambulance and Emergency Care station named after A.S. Puchkov. Unfortunately, we could find no published statistics on gender and age of hospitalized COVID-19 patients in Russia, but in our opinion, patients enrolled in this study reflect the general population of Moscow.

The study was approved by the local ethics committee by the Federal Center of Brain research and Neurotechnologies (project RF-COVID, approved on 5 July 2021). The study was conducted in accordance with the Declaration of Helsinki Ethical Principles. Upon admission, all patients included signed an informed consent that their data may be anonymized and used for aims of scientific research and publication.

### 2.2. Methods of Examination

Within the standard protocol of examination at admission, blood saturation was determined in patients using a pulse oximeter MD300 C2 (Beijing Choice Electronic Tech Co., Ltd., Beijing, China), blood was withdrawn for standard clinical and biochemical analysis with a determination of the complete blood count (CBC), erythrocyte sedimentation rate (ESR), and hematocrit. Transaminases, lactate dehydrogenase, total protein, glucose, creatinine, urea, total cholesterol, and C-reactive protein were measured in all patients. During the hospital treatment process, the blood tests were determined by clinical necessity.

Depending on the severity of the clinical condition, patients were assigned to the wards with the possibility of oxygen therapy (the need for oxygen therapy was 56.3%), or to intensive care units for artificial ventilation (the need for mechanical ventilation was 9.4%) and correction of the patients’ condition. Patients were treated following the constantly updated recommendations of the Ministry of Health.

According to the indications, transthoracic echocardiography was performed on a Philips Epiq 7G ultrasound scanner (Philips Ultrasound, Inc., Bothell, WA, USA) using a sector multi-frequency matrix sensor with a frequency of 1–5 MHz according to the standard protocol with the registration of the end-systolic and end-diastolic volume, stroke volume, Simpson ejection fraction, and systolic pressure in the pulmonary artery.

At admission and during treatment, patients underwent chest CT scans (Optima CT660, General Electric, Boston, MA, USA), and all CT scans for each patient were divided into three groups—performed at the beginning, in the middle (in dynamics), and end of admission (last CT performed in chronological order). The CT protocol consisted of scanning the patient’s chest at the height of maximum inspiration with a voltage of 120 kV and a tube current of 100–350 mA. Image reconstruction was performed with a slice thickness of 0.625 mm. In some cases, when a patient was admitted from an outpatient CT center, the results of the CT study provided by the patient were used. A total of 1519 CT scans were included in the analysis (from 1 to 7 studies per patient, an average of 3.5 per patient). The analysis of the studies was carried out by experienced radiologists of the Federal Center of Brain Research and Neurotechnologies, who received special training.

The probability of viral etiology of the lesion was estimated by the system similar to RSNA recommendations [44], which is based on the visual assessment of characteristic features. However, in the system used there were four categories: no signs of viral pneumonia, and low, medium, and high probability of viral etiology. The severity of the lung damage was determined by the CT0-CT4 scale, which was recommended by the health authorities in Moscow to unify the interpretation of studies and later confirmed by the temporary recommendations of the Russian Society of Radiology (RSR) [45,46] and by summing up the score of the severity of the lesion for each segment of the lungs on the scale CT-SS [47].

The prevailing types of changes, the condition of other organs of the chest and upper abdomen were evaluated. The predominant types of changes (GGO, “crazy paving”, consolidation, curvilinear opacities, and reticular changes) were evaluated according to a self-produced scale, assigning each of the signs a score from 0 to 3, depending on the visual severity. In the subsequent analysis, the scores were converted to percentages as follows: the scores were summed up and the percentage contribution of each feature to the total score was determined.

The distribution of lesion foci was defined as peripheral, peribronchovascular (central), or mixed. The presence of emphysematous changes, lymphadenopathy, pleural effusion, changes in the trachea, the degree of expansion of the aortic root and pulmonary trunk according to CT data, and the presence of calcified plaques were also evaluated.

### 2.3. Statistical Methods

Statistical analysis was performed using the software packages SPSS Statistics version 23.0 (IBM, Armonk, NY, USA) and R software version 3.3.2 (R Core team, Vienna, Austria) [48]. The null hypothesis was rejected at a significance level of *p* < 0.05. For the description of scale variables, the arithmetic means and standard deviation were used, unless otherwise specified, for nominal variables, the frequency and proportion (in percent) were used. The distribution of scale variables was assessed by frequency histograms evaluation. For nominal dependent variables frequencies comparison between categories of independent (grouping) variables were performed using the criterion χ^2^ Pearson or the exact Fischer criterion. For scale dependent variables, analysis was made using ANOVA followed by pairwise comparisons using the Dunnet method or (if the distribution of the variable did not match the normal one)—Kruskal–Wallis criteria with pairwise comparisons by the Mann–Whitney criterion (using Bonferroni multiplicity adjustment). The effect of individual independent variables on patient survival was evaluated using the Kaplan–Mayer log-rank criterion. After identifying potential confounders and predictors using the methods described above, the exact identification of variables that affect the unfavorable outcome probability of coronavirus infection was performed using binary logistic regression.

## 3. Results and Discussion

### 3.1. Epidemiologic Risk Factors

In our study, an unfavorable prognosis was recognized as a fatal outcome, so the main goal of the study was to predict intrahospital mortality.

Deceased patients were significantly older than those who recovered (mean age 69.08 ± 13.50 vs. 58.20 ± 14.93, *p* < 0.0005), that corresponds to results of other studies [15,19,20]. As in other studies, men also had a higher risk of death—OR 1.97 (95% CI 1.07–3.62, *p* = 0.030). This is also consistent with data from other large studies with reasonable variation in risk [3,15]. Due to the revealed heterogeneities, the analysis was further adjusted for gender and age.

When adjusting for age, the difference in mortality among men and women over and under 59 years old is different. Risk of death in males over 59 years old is slightly, but insignificantly, higher than in women, OR 1.83 (95% CI 0.92–3.62), *p* = 0.081. Among people with an age below the median (59 years), men were at higher risk of death (in that group 12 out of 13 registered cases of death were male). OR of death in males under 59 years old is 8.32 (95% CI 1.07–64.75), *p* = 0.018. This observation reproduces data from previously published studies [3].

In contrast to the results of other published studies, patients did not significantly differ in body mass index (29.60 ± 6.12 among those who recovered versus 27.99 ± 5.11 among those who died, *p* = 0.104), which contradicts the data of broader studies. At the same time, the risk of death non-linearly depends on the body mass index and the lowest risk is noted at the level of BMI = 23.7, increasing at the level above and below this value [49], so the curve is J-shaped, and the average BMI in our study, both for survivors and deceased patients, lies in a much higher zone. This can be explained by the fact that a high BMI is associated with a higher risk of hospitalization, and during selection, our in-hospital study included overweight patients with a higher frequency. In addition, an increase in intrahospital mortality was observed with a BMI of more than 30 kg/m^2^ [50] or 35 kg/m^2^ [51], that is, in different classes of obesity, but there is evidence that being overweight (and not obese) does not affect the mortality of patients with coronavirus [52]. There is also the “obesity paradox”, which shows that obese individuals have a lower mortality rate for various diseases, including pneumonia [53], which may be caused by both a higher level of attention from the medical staff and some other factors. If the limit value between overweight and normal weight is consistently set at 25, 30, 35, and 40 kg/m^2^ (according to obesity classes I, II, and III) there was no significant risk for overweight patients in our study.

Among the two groups of patients studied, the frequency of smoking did not significantly differ (8.9% in recovered group vs. 10.2% in deceased group, *p* = 0.798), but in our study, only the fact of current smoking was recorded, without taking into account the duration and intensity of smoking, which could affect the analysis result and lead to a contradiction with the available data [22]. In addition, the same relationship between smoking and the risk of severe viral pneumonia and mortality, as in our study, has been demonstrated before [23]. Even age-adjusted, smoking status had no significance as a risk factor among our patients.

The patients who died were slightly more likely to drink excessive alcohol (5.9% vs. 0.9%, *p* = 0.021), OR was 7.13 (95% CI 1.65–30.72, *p* = 0.0084), which is consistent with the data of previous studies [54], however, there is a wide variability of parameters related to the frequency and intensity of alcohol consumption and that requires careful data interpretation and further research, although a high risk of ARDS among long-term alcohol users has been demonstrated even before the pandemic began [55].

Excessive alcohol consumption, as reported by patients, preserves its prognostic value in different age groups (OR 14.18, 95% CI 2.34–85.90, *p* = 0.0039 in the younger age group), and if age-adjusted this parameter becomes more significant. Self-reported excessive alcohol consumption is not a reliable indicator because patients tend to report less amounts of alcohol consumption. The noteworthy, negative influence of high alcohol consumption on the frequency of ARDS in patients with coronavirus pneumonia was reported earlier [56], and its mechanism was probably realized through the dysfunction of the alveolar epithelium, alcohol-induced oxidative stress, and dysfunction of alveolar macrophages [57].

The main results of epidemiological risk factors analysis are shown in the Table 2.

### 3.2. Comorbidities

Arterial hypertension did not affect survival in the overall analysis of the sample (survivability log-rank test, *p* = 0.33), however, when adjusted for age, patients with arterial hypertension had a significantly better prognosis than patients without it in the age group older than 59 years (OR = 0.37, 95% CI 0.19–0.73, *p* = 0.003). This unexpected result was also noted in the large-scale epidemiological study E. Williamson et al. [58], which reported no effect of hypertension on the risk of death in the general sample, as well as its positive impact on survival in the older age group (more than 70 years) after applying age adjustments. Studies of this phenomenon require an in-depth study of the mutual influence of many factors, including concomitant therapy of patients.

The survival rate was analyzed both in the general population and in groups younger and older than 59 years. Arterial hypertension did not influence the risk of death in coronavirus in the young cohort in our study (patients under median age of 59 years were regarded as young in our study), while in elder patients arterial hypertension was a protective factor. The Kaplan–Meyer analysis of survival in the older age group of patients with or without arterial hypertension is presented on Figure 1.

Thus, the presence of arterial hypertension significantly reduces the chances of dying from coronavirus in the survival analysis. This finding may be linked with the higher frequency of observation of previously diagnosed arterial hypertension, and, accordingly, the longer duration of its treatment. However, there is mixed evidence that common antihypertensive drugs (ACE inhibitors, angiotensin receptor blockers, calcium channel blockers, and beta-blockers) may have a neutral or positive effect in case of coronavirus infection [59].

The presence of a previous myocardial infarction (18.9% in the group of deceased versus 6.9% in the group of survivors, *p* = 0.005) significantly worsened the prognosis of the outcome of the disease. The odds ratio of death is 3.15 (95% CI 1.47–6.73). Similarly, the prognosis of the disease is affected by a history of stroke (11.3% in the fatal group versus 3.3% in the recovered group, *p* = 0.013), OR = 3.78 (95% CI 1.44–9.92). The presence of cancer significantly increased the risk of an unfavorable outcome, it occurred in 13.2% of the deceased versus 4.3% of the recovered patients, *p* = 0.012. The ratio of the chances of death was found to be 3.39 (95% CI 1.39–8.26).

After age adjustment previous MI, stroke, and oncologic diseases had a negative effect on patients’ survival (Mantel-Haenszel common odds ratio estimate, OR = 2.14, 95% CI 0.97–4.7, *p* = 0.06; OR = 2.67, 95% CI 1.01–7.11, *p* = 0.048; OR = 2.74, 95% CI 1.10–6.82, *p* = 0.03, respectively). COPD and bronchial asthma were not found to be significant risk factors of mortality after age adjustment, as well as other diseases.

The worse prognosis in the case of concomitant diseases is due to both the general deterioration of the patient’s condition and some specific effects of these diseases on the outcome of viral pneumonia caused by a coronavirus, e.g., a transmitted coronavirus infection causes direct damage to the myocardium because of hemodynamic disorders or hypoxemia, concomitant myocarditis, stress cardiomyopathy, microvascular damage, and the development of thrombosis due to hypercoagulation or systemic inflammation (cytokine storm). Hypercoagulation, hypoxemia, and systemic inflammation can also be destabilizing coronary artery plaques, rendering them vulnerable, which can provoke vascular catastrophes in both the cerebral and coronary bed, which will be more difficult to tolerate due to the presence of previous damage to the brain and myocardium, respectively.

Data from various sources show that the probability of a serious illness and death COVID-19 is higher in adult patients with cancer, especially with hematological diseases and lung cancer, as well as in the elderly and other concomitant diseases [58,60,61]. This may be due to various reasons, including ongoing chemoradiotherapy, which leads to a decrease in the number of patients treated and the rate of appearance of formed blood elements, including red blood cells necessary for oxygen transport in hypoxia, and white blood cells involved in the formation of an immune response.

The presence of bronchial asthma (3.8% vs. 2.7%, *p* = 0.656), atrial fibrillation (13.2% vs. 8.1%, *p* = 0.189), diabetes mellitus (11.3% vs. 15.6%, *p* = 0.438), viral hepatitis, thyroid diseases, and systemic autoimmune diseases were also not associated with more frequent deaths.

Summary of main mentioned above comorbidities, regarded as a risk factor, is shown in Table 3.

### 3.3. Complications

In the deceased patients, the complication rate was many times higher, but the presence of complications did not mean an unambiguous fatal outcome.

DIC syndrome was observed in 7 patients with fatal outcomes (13.2%) and in 7 recovered patients (1.2%), OR of death among the patient groups was 12.50 (95% CI 4.20–37.17, *p* = 0.0001). Sepsis, by the groups, was observed in 29 and 3 patients, respectively (54.7% and 0.5%), OR of death was 233.20 (95% CI 66.36–819.58, *p* = 0.0001). Deep venous thrombosis was detected in 15 and 3 patients, respectively (28.3 vs. 0.5%), OR 76.18 (95% CI 21.13–274.64, *p* = 0.0001).

Pulmonary embolism during treatment occurred in 13 dead patients and 5 recovered patients (24.5% and 0.9%, respectively), OR 37.5 (95% CI 12.74–110.45, *p* = 0.0001).

The development of ARDS was an unfavorable prognostic sign: all 25 patients with this complication died (47.2% of the patients who died).

Acute myocardial infarction occurred only in 3 patients with a fatal outcome (5.7%), as well as pneumothorax/pneumomediastinum (3 patients, 5.7%).

Registered hemorrhagic complications were not associated with mortality (1 patient in the group of deaths and 2 patients in the group of survivors, *p* = 0.230).

### 3.4. Laboratory Testing

The deceased patients had significantly lower hemoglobin levels (130.10 vs. 139.11, *p* = 0.001; average values are indicated hereafter), red blood cell count (4.26 vs. 4.67, *p* < 0.0005), and hematocrit (39.03 vs. 41.68, *p* < 0.0005). Given that with the development of prolonged respiratory failure, the number of red blood cells increases as compensation (due to a decrease in the concentration of oxygen in the kidneys and an increase in the production of erythropoietin), this phenomenon is not quite usual. This observation is generally consistent with the literature data—anemia in coronavirus infection is often observed and correlates with unfavorable outcomes [62].

The explanation for this phenomenon lies in a combination of general hypoxia (including bone marrow hypoxia) and a systemic inflammatory reaction that reduces the lifetime of red blood cells. A decrease in the level of hemoglobin should worsen hypoxia due to respiratory failure, it is also possible that the level of hemoglobin (or the number of RBC), and the dynamics of this indicator may be a good predictor of an unfavorable outcome, testing this hypothesis, however, requires a separate study.

The leukocytes’ count was significantly higher in patients who later died (8.38 vs. 6.34, *p* < 0.0005), and the number of band neutrophils were also higher (12.43 vs. 5.36, *p* < 0.0005), as well as the count of segmented neutrophils (not significant, 66.71 vs. 58.50). It is worth noting that this trend could be associated with the addition of a bacterial infection.

On the other hand, the number of platelets and lymphocytes is slightly lower (also non-significant—161.41 vs. 181.76 for platelets, *p* = 0.073; 8.69 vs. 12.14 for lymphocytes, *p* = 0.111). The ESR level, the number of monocytes, and eosinophils did not differ between groups.

The ratio of neutrophils to lymphocytes was calculated retrospectively, considering the published data. This criterion turned out to be prognostically significant: with a value of more than 2.5, the survival rate of patients was much lower, as it is presented in Figure 2.

Among patients in the recovered group, the median of this ratio was also lower than in the group of patients with a fatal outcome—2.13 [IQR 0.97; 3.76], and in those who died-4.19 [IQR 1.04; 3.91]. These results are graphically presented in Figure 3.

In the biochemical analysis of blood, it was noteworthy that patients with a fatal outcome had lower levels of total protein (64.74 vs. 71.00, *p* < 0.0005), as well as higher levels of blood glucose (7.61 vs. 6.93, *p* = 0.046), creatinine (94.00 vs. 85.03, *p* = 0.025), and C-reactive protein (153.02 vs. 69.57, *p* < 0.0005). The levels of procalcitonin, a marker of bacterial infection, were not significantly different between groups (1.34 vs. 0.13, *p* = 0.117).

In quantitative blood tests, it was noted that the significance of high creatinine and glucose levels does not affect the outcome (85.0 vs. 94.0, *p* = 0.179 and 6.9 vs. 7.6, *p* = 0.130, respectively). This may be explained by higher levels of glucose and creatinine in the elder population.

Among the main vital signs, a statistically significant difference was found in the frequency of respiratory movements (25.63 in those who subsequently died, 22.04 in those who recovered, *p* = 0.001). Blood pressure, body temperature, and heart rate were almost identical.

### 3.5. Echocardiography

According to echocardiography data, there was a tendency to decrease the ejection fraction and stroke volume in the group of fatal outcomes versus the group of recovered patients (57.56 vs. 60.53, *p* = 0.055 and 64.38 vs. 73.73, *p* = 0.043, respectively), but these values lie in the region of borderline statistical significance. There is a dramatic difference in systolic pressure in the pulmonary artery (53.00 in deceased patients versus 37.61 in recovered patients, *p* < 0.0005) that may be caused by an increase in pulmonary vascular resistance due to endothelitis and pulmonary embolism. Careful analysis of this indicator is required, together with analysis of the functional state of the right ventricle and other modalities for assessing the risk of death and thromboembolic complications.

### 3.6. Radiology

CT scans at every time point (for the initial study, CT in dynamics, and at the last CT scan) demonstrated “crazy paving” as the most pronounced type of changes in the deceased patients, the median value of severity of this type of changes was 22.2% [IQR 0.0–37.5], and in the group of surviving patients—0% [IQR 0.0–25.0] (*p* < 0.0005).

GGO on primary CT were less typical for deceased patients than for those who recovered—37.5 [IQR 20.0–50.0] vs. 42.9 [IQR 25.0–60.0] (*p* = 0.007). However, this shift can be explained by the fact that severe patients, admitted at later stages of the disease, when GGO transforms into zones of consolidation, were more likely to die. This is confirmed by the increase in the severity of the consolidation and “crazy paving” pattern in these patients.

On dynamic CT, patients with a fatal outcome had less prevalence of curvilinear consolidations 0.0 [IQR 0.0–11.1] and reticular changes 5.0 [0.0–15.5] than the surviving patients (14.3 [IQR 0.0–20.0] and 14.3 [IQR 0.0–25.0], *p* = 0.003 and *p* = 0.018, respectively). Probably, these changes are observed in the resolution stage of pneumonia, which was not achieved by deceased patients due to the development of ARDS, or massive consolidations. On the last CT scan performed, “crazy paving” also prevails in deceased patients, reticular changes are less pronounced 11.8 [IQR 0.0–22.2] vs. 25.0 [IQR 0.0–33.3] in survivors (*p* = 0.001) and curvilinear consolidations 9.1 [IQR 0.0–14.3] vs. 16.7 [IQR 0.0–28.6] in survivors (*p* = 0.001). At the same time, consolidations in deceased patients are more pronounced—29.3 [IQR 18.2–37.5] versus 16.7 [IQR 0.0–33.3] (*p* = 0.004). The “crazy paving” pattern is associated with filling the alveoli with exudate and associated thickening of the interlobular and intralobular interstitium, which creates linear strands over the GGO [63]. There are studies in which the appearance of “crazy paving” was assessed as a risk factor, but in the paper of Q. Lei et al. [64] this pattern was not identified as significant. On the other hand, there are data that this pattern can be used as a predictive one [65], and this is confirmed, among other things, by the results of our analysis.

With CT at all time points, both on the sum of points and the CT0–CT4 scale, the changes were more pronounced in severe patients (in all cases, *p* < 0.0005), and both methods of semiquantitative assessment of lung parenchyma involvement resulted in similar risk estimation.

CT changes that were more characteristic for the deceased patients included the measured diameter of the pulmonary trunk, which was significantly larger at all CT scans in these patients than in those who recovered (median 30 [IQR 28.0–34.0] mm vs. 28 [IQR 26.0–30.0] mm at the first CT scan, 32 [IQR 29.0–33.0] vs. 28 [IQR 26.0–30.0] on dynamic CT, 30 [IQR 28.0–34.0] vs. 28 [IQR 26.0–30.0] on the last CT, in every case *p* < 0.0005).

Patients with a severe course also had a predominant mixed and diffuse location of the GGO and zones of consolidation, not limited to the peripheral distribution, as was the case in recovered patients (*p* = 0.0001).

In patients with fatal outcome, hydrothorax was recorded more often than in recovered: at the first CT scan, OR = 3.10, 95% CI 1.34–7.16, *p* = 0.0082, with CT in the dynamics of OR = 3.26, 95% CI 1.35–7.89, *p* = 0.0087, and at the last CT scan OR = 7.09, 95% CI 3.43–14.7, *p* = 0.0001. The increased frequency of this syndrome is probably linked with the development of pulmonary edema and ARDS in the final stages of the disease with an unfavorable outcome.

Lymphadenopathy with enlarged lymph nodes over 10 mm along the short axis was slightly more common in the group of patients with a fatal outcome during primary CT (9.6% had multiple enlarged lymph nodes, 25.0%—single enlarged lymph node, 65.4%—had no lymphadenopathy) than in the group of patients who recovered (1.5%, 15.9%, 82.7%). Similar changes became more pronounced in CT during treatment (16.0% had multiple enlarged lymph nodes, 32.0%—single, 52.0%—without lymphadenopathy) than in the group of patients who recovered (2.9%, 15.3%, 81.8%). The proportion of patients with lymphadenopathy of varying severity increased slightly at the last CT scan performed in the group of fatal outcomes (13.9%, 41.7%, 44.4%), in the recovered group, the distribution was 0.9%, 17.8%, and 81.3%. It is worth noting that in this case, the last CT scan performed for recovered patients was performed shortly before discharge, while for deceased patients, the last CT scan was performed at the height of the disease. Reactive inflammatory changes in the lymph nodes can be observed in patients without severe lung damage [66].

Cardiomegaly was more common among the deceased patients’ group. Thus, at the first CT scan, 32.7% of this group had an increase in the size of the heart according to CT data, while in the group of recovered patients it was 16.9% (*p* = 0.005). When analyzing the obtained data, an OR of 2.38 was found (95% CI 1.28–4.43, *p* = 0.0062). In the dynamic CT study, this ratio did not change significantly (36.0% and 17.9%, respectively), OR 2.57 (95% CI 1.07–6.16, *p* = 0.034). At the last CT scan, 36.1% of the deceased patients’ group had an increase in the size of the heart according to CT data, while in the group of recovered patients it was 18.4% (*p* = 0.011). When analyzing the obtained data, OR 2.50 was found (95% CI 1.21–5.15, *p* = 0.0128). Thus, there was no increase in the number of cases of cardiomegaly during treatment, but it was a risk factor of mortality.

Calcified atherosclerotic plaques in the aorta can become a matrix for the formation of thrombi in the event of a violation of the integrity of their capsule, which can cause vascular thromboembolism. At primary CT, aortocalcinosis was observed in 78.8% of patients in the group of a fatal outcome and 46.9% of patients in the recovered group (*p* = 0.0001). The odds ratio was 4.22 (95% CI 2.13–8.40, *p* = 0.0001). Similar ratios were maintained for dynamic CT (52.4% and 24.0%, *p* = 0.007) and final CT (54.6% and 25.0%, respectively, *p* = 0.001). Similarly, coronary arteriosclerosis was detected, prevailing in the group of patients with a fatal outcome in primary CT (75.0%) versus 41.8% in the group of surviving patients. The odds ratio, in this case, was 4.84 (95% CI 2.23–10.50, *p* = 0.0001).

Thus, the presence of calcified plaques can be a predictor of the fatal outcome of coronavirus infection, which reflects the conversion of plaques into a vulnerable state on the hypoxemic background, a systemic inflammatory reaction, and a general metabolic disorder.

The significance of qualitative signs on CT scans does not change after age adjustment, including the remaining significance of aortocalcinosis as a risk factor of fatal outcomes.

### 3.7. Overall Regression Model

After pairwise comparison most significant risk factors were added to the regression model.

In our model of logit regression with Wald test following parameters were included: protein level in biochemical blood test (OR 0.888 [95% CI 0.825–0.956], *p* = 0.002), pulmonary trunk diameter as it was shown by CT (OR 1.179 [95% CI 1.049–1.326], *p* = 0.006), “crazy paving” severity on CT (OR 1.028 [95% CI 1.007–1.048], *p* = 0.008), summed severity score of lung involvement (OR 1.166 [95% CI 1.087–1.250], *p* < 0.0005), history of arterial hypertension (OR 0.384, [95% CI 0.157–0.940], *p* = 0.012), history of MI (OR 4.248, [95% CI 1.310–13.768], *p* = 0.024), and history of stroke (OR 8.986, [95% CI 1.701–47.464], *p* = 0.016). Pseudo R-squared for overall model was found to be 0.445, Hosmer–Lemeshow goodness of fit test showed *p* = 0.673.

The ROC curve for the summary predictive model is shown in Figure 4, and the final results table are provided in Table 4. The area under the curve is 0.890 (95% CI 0.833–0.948), which corresponds to excellent model quality.

The severity of lung involvement on CT scans and severity of lung injury patterns indicates lung injury, characteristic of severe disease course, causing hypoxemia and systemic inflammatory reaction. Dysfunction in coagulation factors may cause microthrombosis in lung vessels, which aggravates gas exchange in the lungs.

History of previous cardiovascular events, such as MI and stroke, and coronaroaortocalcinosis indicates that vessel walls are already vulnerable and local disturbances in partial oxygen pressure and coagulation system in the blood leads to a higher frequency of complications and death in these patients. The protective effect of arterial hypertension is probably linked with the aforementioned antihypertensive medical therapy and higher attention of medical staff to elder patients.

## 4. Conclusions

In our analysis, data from other studies were partially confirmed. Patient mortality is increased with age, male gender, history of MI, stroke, and oncological diseases. On the other hand, arterial hypertension has a protective effect in case of coronavirus infection against mortality in hospitalized patients, especially in the elder group of patients, that may be linked with medical therapy for that condition.

When evaluating blood tests, attention should be drawn to the neutrophil–leukocyte ratio, which has predictive properties in relation to in-hospital mortality at a value above 2.5. Anemia and lymphopenia were more common in deceased patients.

According to CT data, CT scans at any time point, including those performed upon admission of the patient, had a predictive value, which is of great diagnostic value. A particularly significant risk factor is the diffuse location of the lesions, as well as the “crazy paving” appearance associated with thickening of the pulmonary interstitium, most likely on the background of edema.

It is worth noting that our study had its limitations because not the entire population of infected patients was studied, but exclusively those hospitalized in our tertiary non-specialized center—patients with a clinically more pronounced course of coronavirus infection. The results of the work can be used in the triage of hospitalized patients, including to determine those who need intensive oxygen therapy in connection with concomitant diseases.

## Figures and Tables

**Figure 1 diagnostics-11-01687-f001:**
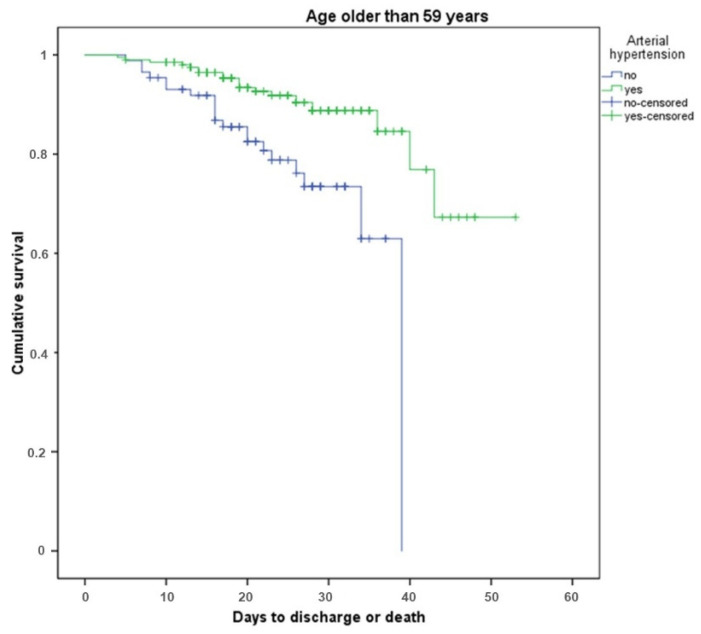
Kaplan–Meyer analysis of survival in age group older than 59 years in patients with and without arterial hypertension. Green line represents patients with arterial hypertension, blue line—patients without arterial hypertension.

**Figure 2 diagnostics-11-01687-f002:**
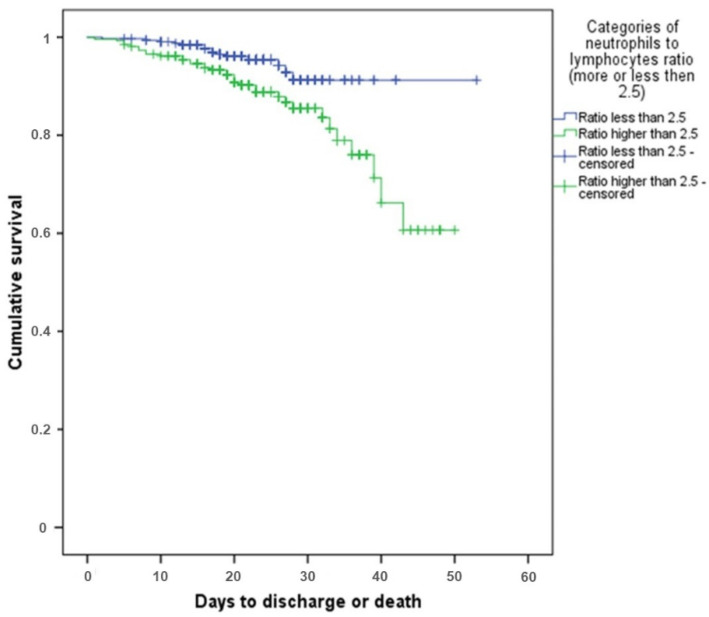
Survival rate in patients’ groups depending on NLR. Blue line represents survival of patients with NLR less or equal than 2.5, green line—patients with NLR more than 2.5.

**Figure 3 diagnostics-11-01687-f003:**
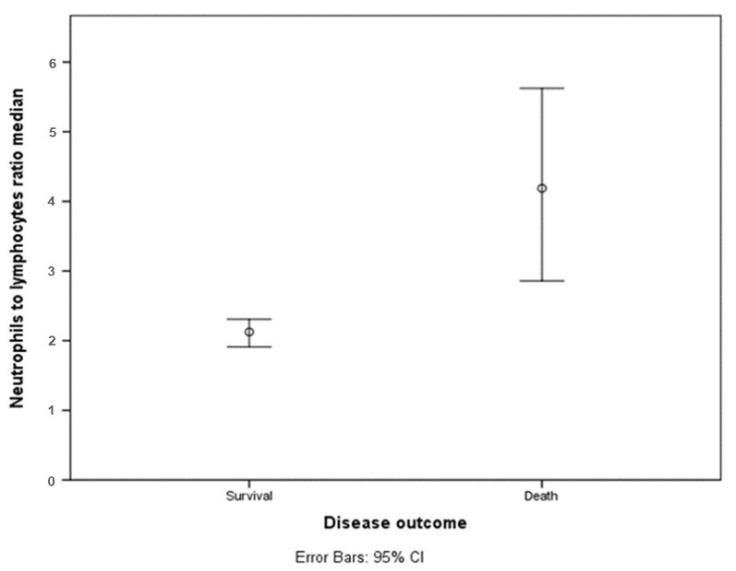
Median and CI of NLR in survivors and deceased patients. NLR is markedly higher in deceased patients’ group. Median is represented with circle; error bars show 95% CI.

**Figure 4 diagnostics-11-01687-f004:**
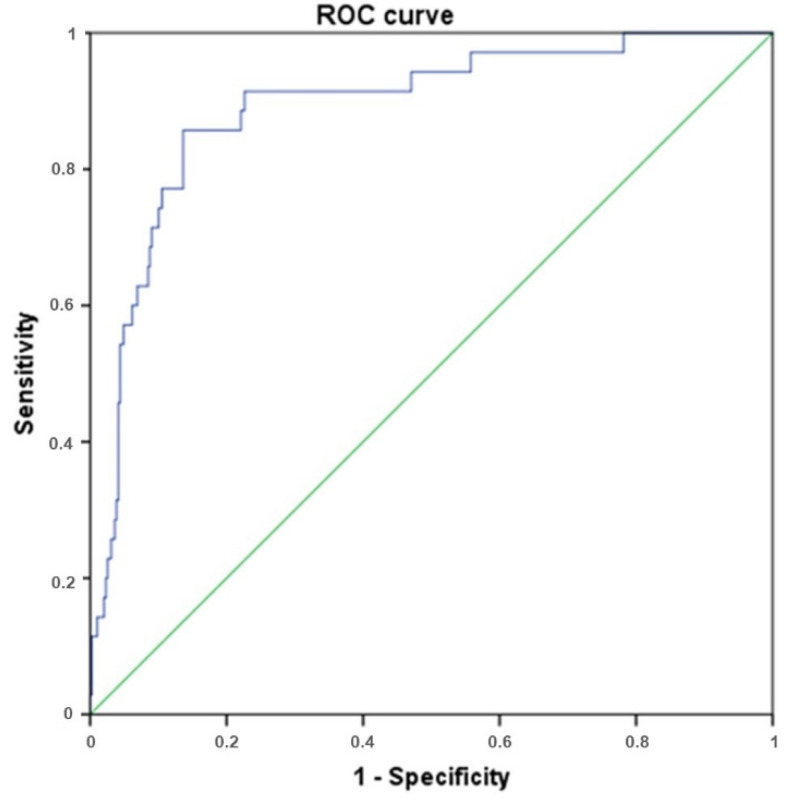
The ROC curve for the overall regression model. AUC is 0.89, which corresponds to the excellent quality of the model.

**Table 1 diagnostics-11-01687-t001:** Summary of the main risk factors of COVID-19 severe outcome or death reported in the literature.

Risk Factor		N	Effect Size	Reference
Gender (male)		n/a 53 countries	FRR 1.35 ^†^ (95% CI 1.35–1.35)	[3]
		16,422 *	FRR 1.45 ^†^ (95% CI 1.23–1.71)	[15]
Age		470,034	65–74 years: RR 4.27 ^†^ (95% CI 3.03–6.03) Over 75 years: RR 12.77 ^†^ (95% CI 9.13–17.85)	[21]
Smoking		1549 *	RR = 2.4 ^‡^ (95% CI: 1.43–4.04)	[22]
		1399 *	OR 1.69 ^‡^ (95% CI 0.41–6.92)	[23]
		369,287 *	OR 1.35 ^†^ (95% CI 1.12–1.62)	[24]
Body mass index		12,591 *	0.89 ^†^ (95% CI 0.32–2.51)	[25]
		7606	OR 1.26 ^†^ (95% CI 1.00–1.58)	[26]
Comorbidities	cerebrovascular diseases	6270 *	OR 4.85 ^‡^ (95% CI 3.11–7.57)	[22]
	cardiovascular diseases		OR 4.81 ^‡^ (95% CI 3.43–6.74)	
	chronic lung diseases		OR 4.19 ^‡^ (95% CI 2.84–6.19)	
	malignancies		OR 3.18 ^‡^ (95% CI 2.09–4.82)	
	diabetes		OR 2.61 ^‡^ (95% CI 2.02–3.3)	
	arterial hypertension		OR 2.37 ^‡^ (95% CI 1.80–3.13)	
Thrombocytopenia		1779 *	RR 5.1 ^‡^ (95% CI 1.8–14.6)	[30]
NLR (more than 3.3)		93	RR 2.46 ^‡^ (95% CI 1.98–4.57)	[32]
C-reactive protein		5872 *	WAD 42.7 mg/L ^‡^ (95% CI 31.12–54.28)	[34]
ALT			WAD 5.12 U/L ^‡^ (95% CI 0.82–9.42)	
AST			WAD 8.51 U/L ^‡^ (95% CI 5.01–12.01	
Creatinine			WAD 4.57 µmol/L ^‡^ (95% CI 0.64–8.50)	
CT semi-quantitative scale		130	HR 8.33 ^†^ (95% CI 3.19–21.73)	[36]
CT “crazy paving” sign		217	OR 0.661 ^‡^ (95% CI 0.147–2.974)	[38]

N—number of patients in the study, if available; FRR—fatality rate ratio, RR—risk ratio; OR—odds ratio; WMD—weighted mean difference; CI—confidence interval; NLR—neutrophil –to-leucocyte ratio, ALT—alanine aminotransferase, AST—aspartate aminotransferase; CT—computed tomography. Number of patients marked with an asterisk (*) shows overall number of patients included in the meta-analysis. Effect size marked with ^†^ indicates risk of death, while ^‡^ designation stands for risk of the severe course of the disease.

**Table 2 diagnostics-11-01687-t002:** Summary of key findings among epidemiological risk factors.

Risk Factor		N_R_	N_D_	OR
Gender (m vs. f)	Male	314	37	1.97 * (95% CI 1.07–3.62)
	Female	268	16	
Gender (m vs. f, aged 59 or more)	Male	125	25	1.83 (95% CI 0.92–3.62)
	Female	137	15	
Gender (m vs. f, under 59 years old)	Male	189	12	8.32 * (95% CI 1.07–64.75)
	Female	131	1	
Body mass index	<25	124	11	0.78 (95% CI 0.38–1.60)
	≥25	420	29	
Body mass index	<30	323	27	0.70 (95% CI 0.36–1.39)
	≥30	221	13	
Current smoking	Yes	51	5	1.17 (95% CI 0.44–3.08)
	No	525	44	
Alcohol abuse (self-reported)	Yes	5	3	7.13 * (95% CI 1.65–30.72)
	No	570	48	

N_R_—number of patients who had the risk factor in recovered group; N_D_—number of deceased patients who had the risk factor, OR—odds ratio. OR marked with an asterisk (*) are statistically significant.

**Table 3 diagnostics-11-01687-t003:** Summary of key findings in analysis of comorbidities as a risk factor.

Risk Factor		N_R_	N_D_	OR
Arterial hypertension	Yes	294	24	0.81 (95% CI 0.46–1.43)
	No	288	29	
Arterial hypertension (for patients over 59 years)	Yes	191	20	0.37 * (95% CI 0.19–0.73)
	No	71	20	
Arterial hypertension (for patients under 59 years)	Yes	103	4	0.94 (95% CI 0.28–3.11)
	No	217	9	
Myocardial infarction (in history)	Yes	40	10	3.15 * (95% CI 1.48–6.73)
	No	542	43	
Stroke (in history)	Yes	19	6	3.78 * (95% CI 1.44–9.92)
	No	563	47	
Malignancy (in history)	Yes	25	7	3.39 * (95% CI 1.39–8.26)
	No	557	46	

N_R_—number of patients who had the risk factor in recovered group; N_D_—number of deceased patients who had the risk factor, OR—odds ratio. OR marked with asterisk (*) are statistically significant (*p* < 0.05).

**Table 4 diagnostics-11-01687-t004:** The results table for the final overall regression model.

	B	Significance Level	OR	95% CI for OR	
				Low	High
Protein level in biochemical blood test	−0.119	0.002	0.888	0.825	0.956
Pulmonary trunk diameter	0.165	0.006	1.179	1.049	1.326
“Crazy paving” severity on CT	0.027	0.008	1.028	1.007	1.048
Summed severity score of lung involvement	0.153	<0.0005	1.166	1.087	1.250
No history of arterial hypertension	1.422	0.012	4.146	1.371	12.536
No history of MI	−1.564	0.024	0.209	0.054	0.812
No history of stroke	−2.420	0.016	0.089	0.012	0.635
Constant	−0.371	0.913	0.690		

B—value for the logistic regression equation for predicting the dependent variable from the independent variable, OR—odds ratio, CI—confidence interval.

## Data Availability

The datasets generated during and/or analyzed during the current study are available from the corresponding author on reasonable request.

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
