# Peer review of "Risk Factors of In-Hospital Mortality in Non-Specialized Tertiary Center Repurposed for Medical Care to COVID-19 Patients in Russia"

_diagnostics, 2021, doi:10.3390/diagnostics11091687_

Round 1

Reviewer 1 Report

This is an extremely well documented and carefully presented study. The purpose is to analyse risk factors and outcomes in a sample of 635 patients with Covid-19 admitted to a repurposed medical centre and compare to the world literature to date. Therefore, the comparability of the sample and the ease with which comparisons are made to the world literature by readers are key. 

Introduction

Excellent literature review, which prepares the reader to compare your study to the world literature. Can you please make a table using the data already in your text of the literature your report with n's of the included studies so that the reader can comprehend your lit search results easily? Text is heavy reading - at the end you have to go back to the beginning to remember what you read. 

Sample

How was your sample selected for transfer? Please compare the basic characteristics of your sample to Russian averages (if known) for COVID-19 so we can understand how generalisable your results are.

Results

Your analysis is comprehensive. Can you please make another table of your  patients' characteristics (with n's so your sample can be compared to averages and ranges in the world literature if possible) so that readers can easily compare your samples and identify if your sample is average or an outlier? N's for subgroups are important to analyse what weight we can place on subgroup analyses within the 635 patients.

My comments are intended to strengthen your study for wider relevance to readers and more citations. My recommendation is minor revision because the basic paper is very sound.

English language: Excellent! Minor suggestions are below:

48 what is sun crown (sunflower?)

50 a range of R numbers have been cited in the literature as the pandemics spread and especially changed with the variants. Please update.

82 incomplete sentence

86 change to severely ill patients

88 coagulopathy. These change are important. Please expand this description

91 variates to varies 

173-178 please provide units for all labs tests (they differ in Europe and the USA)

188-197 Can you provide n's for the studies and the various outcomes? [small studies may be less generalisable]

267 self produced scale. Please state how your scale differs from others and what it adds.

305-309 for those < 59 12/13 deaths were males. These are small numbers and illustrates the need for a comprehensive table with n's

Overall, well done!

106 metaanalyze to metaanalaysis by

119 basing on to based on

161 ... do the 3 dots mean 57 to -39 

Author Response

Dear Reviewer!

Thank you very much for your professional review and valuable comments. We have made changes according to your comments in the manuscript.

Please, find our answers and comments below.

  1. Q:  "Can you please make a table..." A: The table of previous studies, along with key findings and numbers of involved patients, was added to the Introduction section (table 1).
  2.  Q: How was your sample selected for transfer? A: Patients were transferred to the hospital by ambulance, based on their location and worrying signs, such as low blood oxygen saturation level (below 93) and moderate or severe dyspnea. The additional selection criteria were not applied. The patient routing was provided by Moscow Ambulance and Emergency Care Center. This information was added to the manuscript.
  3. Q: Please compare the basic characteristics of your sample to Russian averages. A: Unfortunately, we are unable to find definitive published results about the basic characteristics of the Russian population infected with the SARS-CoV2 virus. Please, find published results about deceased patients added to the manuscript (right after Table 1). We also added few references to the prevalence of some risk factors in the general Russian population.
  4. Q: Can you please make another table of your patients' characteristics A: The tables, summarizing some of the main findings were added to the manuscript (please, find Tables 2 and 3).
  5. Minor issues:

    48 what is sun crown - changed to a stellar crown, the common English term

    50 a range of R numbers have been cited in the literature - R numbers were updated according to recent reports

    82 incomplete sentence - The sentence was corrected

    86 change to severely ill patients - The sentence was corrected

    88 coagulopathy. These change are important. Please expand this description - Additional information was added about coagulopathy, with a reference to a recent comprehensive review. 

    91 variates to varies - The sentence was corrected

    173-178 please provide units for all labs tests (they differ in Europe and the USA) - The units are now indicated in the test

    188-197 Can you provide n's for the studies and the various outcomes? [small studies may be less generalisable] - N's were indicated in the text.

  6.  

    267 self produced scale. Please state how your scale differs from others and what it adds. - This scale was produced in the department in the early phases of coronavirus pandemics in order to ease data collection. The main difference is the way of documentation for each sign (GGOs, consolidation, and so on) prevalence on a 4-point visual scale (from 0 to 3) for each sign, followed by calculating each factor's contribution to the summed score. 

    305-309 for those < 59 12/13 deaths were males. These are small numbers and illustrates the need for a comprehensive table with n's - The table was provided in the manuscript.

    106 metaanalyze to metaanalaysis by - The sentence was corrected

    119 basing on to based on - The sentence was corrected

    161 ... do the 3 dots mean 57 to -39 - Yes, but the sentence was corrected for better comprehensibility. 

Reviewer 2 Report

The present ms investigates risk factors of in-hospital mortality for COVID-19 patients in Russia. The ms is generally well written.

I have only a few comments:

  1. Abstract line 28: missing parenthesis ”)“
  2. Please provide empirical evidence that persons with missing data (removed in the analysis) are similar to persons without missing data. In general, I would prefer a multiple imputation procedure over listwise deletion.
  3. 278: provide a reference for R
  4. 277ff.: I do not think that the approach of the authors is statistically sound. I strongly argue against the approach of choosing either the mean or the median based on deviations from the normal distribution assumption. The median targets a different population quantity than the mean. Either use the median for all variables in the ms or the mean. Moreover, the same argument applies to the choice ANOVA vs. Kruskal-Wallis. If the median is the quantity of choice, use Kruskal-Wallis. If not, use ANOVA. If the mean should be used and the normal distribution assumption is questioned, use alternative ANOVA inference methods (e.g., resampling or permutation tests) instead.
  5. Always accompany p-values (e.g., l. 329 or l. 342) with corresponding effect sizes (differences in proportions, correlations, …).
  6. l. 569ff.: Present a results table for the final overall regression model.

Author Response

Dear reviewer!

Thank you very much for your expertise and valuable review. 

We made some changes to the manuscript according to your comments.

  1. Abstract line 28: missing parenthesis ”)“ - corrected
  2. Please provide empirical evidence that persons with missing data (removed in the analysis) are similar to persons without missing data - The causes of missing data in our dataset were equipment failures and insufficient experience in report forms filling in by few investigators, especially during the early phase of the first pandemic wave. These causes do not seem to be associated with disease severity, which is confirmed by approximately equal missing frequency in survived and deceased patients.  Thus we suppose that our missing data occur completely at random order, so listwise deletion would not lead to biases. 
    Thank you a lot for your suggestion about the multiple imputation procedure. We will seriously consider it in our subsequent works.
  3. 278: provide a reference for R - the reference provided
  4. 277ff.: I do not think that the approach of the authors is statistically sound. I strongly argue against the approach of choosing either the mean or the median based on deviations from the normal distribution assumption. The median targets a different population quantity than the mean. Either use the median for all variables in the ms or the mean. Moreover, the same argument applies to the choice ANOVA vs. Kruskal-Wallis. If the median is the quantity of choice, use Kruskal-Wallis. If not, use ANOVA. If the mean should be used and the normal distribution assumption is questioned, use alternative ANOVA inference methods (e.g., resampling or permutation tests) instead. - wherever possible, the changes were made and means were used. Medians are still presented in the text for ordinal values. ANOVA was used for all quantitative values.
  5. Always accompany p-values (e.g., l. 329 or l. 342) with corresponding effect sizes (differences in proportions, correlations, …). - corrections were made to the manuscript
  6. l. 569ff.: Present a results table for the final overall regression model. - Table 4 presents results for the final overall regression model.